# Exploration of the Relationships between Men’s Healthy Life Expectancy in Japan and Regional Variables by Integrating Statistical Learning Methods

**DOI:** 10.3390/ijerph20186782

**Published:** 2023-09-19

**Authors:** Fumiya Sato, Keiko Nakamura

**Affiliations:** Department of Global Health Entrepreneurship, Graduate School of Medical and Dental Sciences, Tokyo Medical and Dental University, 1-5-45 Yushima Bunkyo-ku, Tokyo 113-8519, Japan

**Keywords:** linear regression, regression tree, healthy life expectancy, health policy making

## Abstract

A quantitative understanding of the relationship between comprehensive health levels, such as healthy life expectancy and their related factors, through a highly explanatory model is important in both health research and health policy making. In this study, we developed a regression model that combines multiple linear regression and a random forest model, exploring the relationship between men’s healthy life expectancy in Japan and regional variables from open sources at the city level as an illustrative case. Optimization of node-splitting in each decision tree was based on the total mean-squared error of multiple regression models in binary-split child nodes. Variations of standardized partial regression coefficients for each city were obtained as the ensemble of multiple trees and visualized on scatter plots. By considering them, interaction terms with piecewise linear functions were exploratorily introduced into a final multiple regression model. The plots showed that the relationship between the healthy life expectancy and the explanatory variables could differ depending on the cities’ characteristics. The procedure implemented here was suggested as a useful exploratory method for flexibly implementing interactions in multiple regression models while maintaining interpretability.

## 1. Introduction

Measuring health levels and identifying related factors are one of the key roles in health policy making. Among several health indicators, healthy life expectancy (HLE) has been widely used because it provides comprehensive information on the functional health status of populations [1,2,3,4,5]. In Japan, the second term of the national health promotion plan named “Health Japan 21” began in 2013, which aimed to extend HLE as a main goal [6]. There has also been health research on HLE at local levels in Japan suggesting that socio-demographic and other regional factors are associated with HLE, though a comprehensive understanding of factors related to HLE remains to be elucidated due to the complexity arising from its multifactorial nature [7,8]. Based on a multiple linear regression analysis of the 331 secondary medical areas of Japan, it was suggested that regional factors, such as the tax revenue of tobacco and the proportion of elderly single-person households, have a negative association with HLE, whereas those such as the number of dementia supporters and the proportion of graduates with a university degree or higher have a positive association [8].

Linear regression is a versatile tool in health research to quantitatively analyse the relationship between one or more explanatory variables and an outcome variable of interest, e.g., HLE [8,9,10]. While its relative intuitiveness helps researchers and policy-makers understand results, the assumption of simple linearity may not hold true where the relationships between variables are more complex. Adding interaction and/or polynomial terms may be beneficial to account for nonlinearity; however, it should be carefully applied because increasing model complexity can lead to reduced interpretability of the model and/or overfitting. Determining nonlinear terms to include with appropriate functional forms may also be difficult when there is a lack of prior information. These concerns extend to locally weighted regression as well, where selecting an appropriate weight function that is both effective and interpretable can be challenging.

Other ways to extend linear regression concerning nonlinearity include using piecewise models, where partitioning a dataset into pieces and regression thereof may be performed either sequentially, as in M5′, or simultaneously, as in MPTree [11,12,13,14]. The use of decision tree-based splitting combined with linear regression in each leaf enables these approaches to maintain the interpretability of both of the submodels. Single tree models, however, are prone to overfitting and may lack generalization ability; crisp node splitting may not sufficiently reveal relationships between the difference of estimated coefficients in multiple linear regression for each leaf and characteristics of the leaves indicated from conditions for splitting where true clusters have continuous variations. Though the MPTree model constructs a single regression tree by recursive binary partitioning with relatively high prediction accuracy and small numbers of terminal leaf nodes [12], its reliance on the use of polynomial regression functions may make it difficult to intuitively interpret the relationships between variables represented by the model.

To address these challenges, we developed “sylvan linear regression”, which is a novel regression model implemented by combining multiple linear regression and an ensemble of tree-based models known as random forest. Random forest is an ensemble-learning method that builds multiple decision trees on bootstrapped sample sets while randomly selecting a subset of features for node splitting [15,16]. By aggregating predictions of the trees, random forest improves accuracy and stability with reduced overfitting. The capability to capture complex interactions and nonlinear relationships between variables with varying degrees of influence is another strength of random forest. Sylvan linear regression also adopts bootstrap sampling and random feature selection to create a diverse ensemble of decision trees each leaf of which has estimated regression coefficients by node-wise fitting with multiple linear regression, as described in the Section 2. This paper aims to demonstrate the use of this model for exploring heterogeneity of relationships between municipal factors and HLE as a representative case, where changes in regression coefficients of the ensemble calculated for each sample are examined.

## 2. Materials and Methods

HLE of men was calculated as an outcome variable with the HLE calculation program published by a Japanese Ministry of Health, Labour and Welfare research group [17], in which we used municipal lifetime tables and data of the number of people requiring support in the long-term care insurance system, with care level 2 and above being a surrogate of unhealthy status. The mean and standard deviation of the men’s HLE across the cities were 79.39 and 0.77, respectively. We collected statistical data with a base year of 2015 from 544 cities that have a population of at least 50,000, using open data sources of the Japanese government and a specified non-profit organization [18,19,20,21]. We conducted the analysis using all 544 cities for which data were collected. The cities included in our analysis represent approximately 84.2% of the national population of Japan, which had a total of 1741 municipalities in 2015. When surveys as the data source were conducted only every few years (i.e., not annually) and did not cover the base year of 2015, we linearly interpolated data from the nearest years or applied the last observation carried forward if values of the newer year were not provided yet. The list of 16 explanatory variables is shown in Table 1, with their abbreviations which are used in the following sections. The original data sources for each variable are as follows: proportion of graduates with tertiary education or higher (TE), the Population Census (Kokusei Chousa); Proportion of elderly (aged ≥ 65) single-person households, men (ES), the Population Census; unemployment rate (UE), the Population Census; proportion of workers aged 65 and over (EW), the Population Census; proportion of primary industry workers (PW), the Population Census; proportion of primary and secondary industry workers (IW), the Population Census; proportion of car-only commuters (CM), the Population Census; taxable income (TI), the Survey of Municipal Taxation Status (Shichoson Zei Kazei Jokyo To no Shirabe); financial strength index (FI), the Survey of Local Government Finances (Chiho Zaisei Jokyo Chosa); municipal tobacco tax collection (TB), the Survey of Local Government Finances and the Population Census; number of dementia supporters (DS); Ninchisho (Dementia) Supporter Caravan and the Population Census; number of participants in community-based salons for preventive care (CS), the Survey Results on the Long-term Care Prevention Projects and the Comprehensive Projects for Long-term Care Prevention and Daily Life Support (Regional Supporting Projects) (Kaigo Yobo Jigyo oyobi Kaigo Yobo Nichijo Seikatsu Shien Sogo Jigyo (Chiiki Shien Jigyo) no Jisshi Jokyo ni Kansuru Chosa Kekka) and the Population Census; total population (TP), the Population Census; proportion of inhabitable areas (IA), the Municipalities Area Statistics of Japan (Zenkoku Todofuken Shikuchoson Betsu Menseki Cho); population density in inhabitable areas (PD), the Municipalities Area Statistics of Japan and the Population Census; Proportion of population without flush toilet (NT), the Survey on Disposal of General Waste (Ippan Haiki-butsu Shori Jigyo Jittai Chosa). All the variables were standardized for further analysis.

Sylvan linear regression, a regression model proposed in this work, firstly builds multiple trees by recursive binary node splitting of bootstrap sample sets generated by random sampling with replacement (Figure 1a). Given a tree node and a candidate subset of j features randomly selected for node splitting, the model reiteratively finds a single breakpoint of the features for the two child nodes under the conditions as follows (Figure 1b): (1) the sum of the mean-squared errors (MSE) of multiple linear regression for each child node is minimized; (2) the resulting reduction in errors obtained by subtracting the sum from the MSE for the parent node exceeds a certain constant fraction (α) of the MSE for the root node (E_r_); (3) the child node sizes do not fall below a predefined minimum limit (m). The second and third conditions are introduced to avoid overfitting. Multiple regression analyses on each node of the generated trees were conducted by including all 16 variables listed in Table 1 as explanatory variables, regardless of a set of selected variables as features considered for node splitting. After all nodes of all the generated trees have turned into leaves, i.e., splitting has stopped for every tree, regression coefficients of the ensemble are calculated for each sample by taking the means of the regression coefficients of all the single leaves to which each was assigned across the trees (Figure 1a). The procedures are illustrated in Figure 1. The regression model was implemented in Python; the generation of each decision tree was based on the algorithm using stacks [16]. Hyperparameters of the model must be tuned through cross-validation to optimize the MSE of predicted values obtained by averaging predictions from multiple linear regression models of the leaves. We conducted a 20-fold cross-validation with 200 trees for each round.

After the generation of a forest comprising decision trees with bootstrap sample sets for each, we created scatter plots of all the sample cities for each explanatory variable with the averaged regression coefficients on Y-axes to explore potential interactions of the variables. The first principal component (PC1) of the standardized explanatory variables was selected as the X-axes variable under the assumption that plotting with PC1, the single variable indicating the direction with the largest variance of the original dataset, on the X-axes effectively visualizes the heterogeneity of Y-values and possible interactions with explanatory variables. Putative two-way interaction terms suggested by the plot were exploratorily added as additional explanatory variables in the final multiple linear regression model; an interaction term between an explanatory variable x and a specific variable z (PC1 in this case) was not given as the simple product of x and z, but as the product of x and the image of z under one of the following piecewise linear functions with exploratorily defined parameters:(1)$$lins(z;\;a,\;b)=f(z)-f(0),\;\mathrm{where}\;f(z)=\max(\min(\frac{z-a}{b-a},\;1),\;0)$$
(2)$$linz(z;\;a,\;b)=f(z)-f(0),\;\mathrm{where}\;f(z)=\max(\min(\frac{b-z}{b-a},\;1),\;0)$$
(3)$$trpz(z;\;a,\;b,\;c,\;d)=f(z)-f(0),\;\mathrm{where}\;f(z)=\max(\min(\frac{z-a}{b-a},1,\frac{d-z}{d-c}),\;0)$$

The subtraction by f(0) was employed as a centering factor so that interaction terms should vanish when z = 0. The hyperparameters of the functions were manually determined.

## 3. Results

Figure 2 shows scatter plots of standardized linear regression coefficients (β) of explanatory variables on Y-axes against PC1 on X-axes for each data point across all the cities included in the analysis. The regression coefficients of the ensemble were calculated by using the sylvan linear regression model comprising 5000 trees trained with hyperparameters m (minimum node size), j (number of features considered for node splitting), and α (proportion of improvement in errors required for node splitting with respect to the error at the root node) tuned via the aforementioned cross-validation to be 45, 4, and 0.02, respectively. Table 2 summarizes the loadings of PC1, which indicate the contribution of each regional variable to it, showing the characteristics of the variable used for *X*-axes in Figure 2. PC1 had an explained variance ratio of 39% and appeared to be mainly related to urbanization. In each scatter plot in Figure 2, the dispersion of the data points along the β values of the *Y*-axis reflects the potential instability of the estimates in the simple multiple regression model where the weights of the municipalities are equally treated. Among the scatter plots, those without apparent changes in the distribution of the β values along PC1 of the *X*-axis (e.g., ES, FI, IA) may mean that the current model does not sufficiently capture the heterogeneity of the relationships between variables or they are indeed almost homogeneous. On the other hand, the scatter plots that exhibit changes in trends along PC1 (e.g., EW, IW, TI) suggest that there can be higher-order relationships between variables which may be captured by introducing interaction terms.

In Figure 2, three types of lines are also depicted on the scatter plots: red dotted lines are drawn at β = 0; orange dashed lines correspond to single β for each explanatory variable obtained by vanilla multiple linear regression (i.e., without interaction terms) with ordinary least squares as described in Table 3 (adjusted R-squared = 0.60); violet solid lines represent changes of β as functions of PC1 into which interaction terms are incorporated, approximating the data points with piecewise linear functions. The latter two lines were drawn from the data of all the 544 cities by each regression analysis. The suggestively positive relationship between HLE and TE, and the negative ones with ES, UE, and TB in Table 3 were consistent with the previous studies, whereas the relationships with the variables such as TI and PD were not nominally significant in this model [7,8]; in addition, the βs of IW and IA were estimated to be positive and negative, respectively.

Table 4 describes the results of the regression model including interaction terms introduced from the implications from Figure 2, interpretations of which are discussed in the Section 4; the adjusted R-squared was 0.64 in this model. The standardized partial regression coefficients of the original explanatory variables are consistent with Y-values on the violet solid lines in Figure 2 at PC1 equals zero. The absolute β values of the interaction terms are reflected in the vertical distance of the corresponding violet line segments parallel to the X-axes.

## 4. Discussion

Applying sylvan linear regression to the dataset showed multifaceted insights into global and local relationships between HLE and the municipal factors with an improved explanatory power. Unexplained residuals of the objective variable in the regression model may be at least partially attributed to the measurement error of HLE itself in addition to unincorporated factors because the calculation of HLE for cities with small populations can be inaccurate [17].

The overall coefficients of the explanatory variables were expressed as the functions of PC1, which were obtained from the results including the interaction terms in Table 4 and visualized as the violet lines in Figure 2. While these violet lines were largely consistent with the orange dashed lines representing the coefficients as single constants that were calculated regardless of any interaction terms in Table 3, the nonzero coefficients of the interaction terms in Table 4 suggested that the magnitude of the relationships between HLE and the explanatory variables could vary according to PC1. For example, IW was suggested to typically have a positive relationship with HLE; it was also implied, however, that the strength of the relationship between them tended to become smaller with increasing PC1. Thus, the results from multiple linear regression with interaction terms investigated via our proposed method can be directly compared with the usual multiple linear regression concerning regression coefficients themselves as well as their coefficients of determination or predictive performance.

The partial regression coefficients of each explanatory variable itself, which corresponded with overall coefficients at PC1 equals zero, could be calculated as those for other PC1 scores as well. One may adopt f(a) as a subtraction term in the piecewise linear functions introduced in the methods section, where a is a real number other than zero, so that coefficients with PC1 of a and their precisions can be directly estimated. Interactions between the explanatory variables and other variables than PC1 can also be visualized by plotting data points as in Figure 2, which may lead to form alternative regression models. There may be principal components that cover even larger variances than the first principal component generated by linear combinations when nonlinear functions are taken into consideration, though the intuitive interpretation of such principal components as axes used for the exploration of interaction terms in the analysis can be difficult. The use of variables such as kernel principal components may be one of the directions of further research.

The application of the piecewise linear functions for one variable to summarize fluctuations of the coefficients calculated for each sample from the trees allows for exploring models that reconcile flexibility and interpretability. Since the piecewise linear functions can be introduced in flexible combination(s) while maintaining their local linearity, we considered these functions suitable for our purpose to extend the usual multiple regression analysis by treating the partial regression coefficients as functions f(z) with a common variable z. However, given that the curve fittings with piecewise linear functions to the plots are exploratory, i.e., their parameters are manually tuned, the nominal p-values in the multiple regression model with interaction terms should be interpreted with caution. Other forms, such as polynomials or splines, could be employed, albeit at the expense of simplicity. For example, it may be possible to consider higher-order interactions by introducing quadratic or cubic terms; in such cases, however, one may also encounter challenges of complicated and cumbersome parameter tuning of f(z) as well as difficulty in interpreting the results. Further studies would include developing automated methods with prespecified hyperparameters to simultaneously fit the data points of coefficients obtained from the forest and optimize the final multiple regression model, in which variable selection techniques such as Lasso could also be used, as well as establishing a solid theoretical foundation for the sylvan regression model by elucidating its mathematical and statistical properties in detail towards establishing the current method as an analytical approach that can be applied to general situations.

## 5. Conclusions

We demonstrated the utility of the sylvan linear regression procedure through the illustrative exploration of factors related to the healthy life expectancy of men in Japan at the municipal level. By introducing the interaction terms that involve piecewise linear functions suggested from the scatter plots of the first principal components and the calculated coefficients for each city, the final multiple regression model with enhanced explanatory power was obtained while balancing adaptiveness and conciseness. Our newly developed method model may pave the way for tailored, healthy city planning based on deeper insights into the relationships between healthy life expectancy and its related factors.

## Figures and Tables

**Figure 1 ijerph-20-06782-f001:**
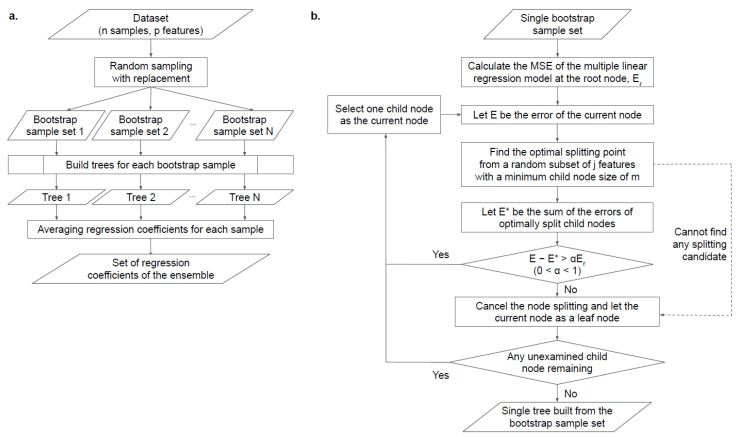
Procedures of sylvan linear regression. (**a**) Construction of a forest consisting of decision trees; (**b**) flowchart of building a single tree.

**Figure 2 ijerph-20-06782-f002:**
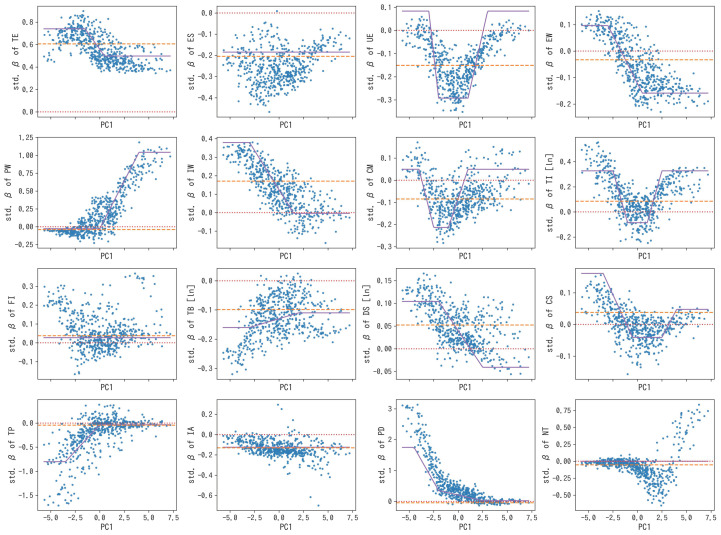
Quantitative relationships between standardized partial regression coefficients (std. β) of the explanatory variables for each city aggregated from the sylvan linear regression model and PC1. Red dotted lines, β = 0; orange dashed lines, estimated β as constants; violet solid lines, approximated β as piecewise linear functions of PC1.

**Table 1 ijerph-20-06782-t001:** List of 16 explanatory variables.

Variable	Unit/Calculation	Abbreviation
Proportion of graduates with tertiary education or higher	%	TE
Proportion of elderly (aged ≥ 65) single-person households, men	%	ES
Unemployment rate	%	UE
Proportion of workers aged 65 and over	%	EW
Proportion of primary industry workers	%	PW
Proportion of primary and secondary industry workers	%	IW
Proportion of car-only commuters	%	CM
Taxable income	yen, per taxpayer, ln-transformed	TI
Financial strength index	basic financial revenue divided by basic financial need	FI
Municipal tobacco tax collection	thousand yen, per thousand people aged ≥ 15, ln-transformed	TB
Number of dementia supporters	per thousand people aged ≥ 65, ln-transformed	DS
Number of participants in community-based salons for preventive care	per thousand people aged ≥ 65	CS
Total population	person	TP
Proportion of inhabitable areas	%	IA
Population density in inhabitable areas	person, per hectare	PD
Proportion of population without flush toilet	%	NT

**Table 2 ijerph-20-06782-t002:** Loadings of the first principal component of the explanatory variables.

Variable	Loading for PC1	Variable	Loading for PC1
TE	0.84	FI	0.63
ES	0.59	TB	−0.056
UE	−0.18	DS	−0.21
EW	−0.59	CS	−0.24
PW	−0.78	TP	0.38
IW	−0.75	IA	0.66
CM	−0.87	PD	0.81
TI	0.85	NT	−0.65

**Table 3 ijerph-20-06782-t003:** Partial regression coefficients of multiple linear regression without interaction terms.

	Std. β	Std. Error	*p*-Value	95% CI	
const.	0	0.027	1	−0.053	0.053
TE	0.6056	0.071	<0.001	0.465	0.746
ES	−0.2048	0.058	<0.001	−0.319	−0.091
UE	−0.1508	0.036	<0.001	−0.222	−0.079
EW	−0.0328	0.04	0.413	−0.111	0.046
PW	−0.0384	0.046	0.405	−0.129	0.052
IW	0.1705	0.052	0.001	0.069	0.272
CM	−0.0849	0.06	0.16	−0.203	0.034
TI [ln]	0.0847	0.068	0.214	−0.049	0.218
FI	0.0378	0.049	0.444	−0.059	0.135
TB [ln]	−0.099	0.038	0.01	−0.174	−0.024
DS [ln]	0.0524	0.029	0.075	−0.005	0.11
CS	0.0379	0.028	0.177	−0.017	0.093
TP	−0.0451	0.03	0.133	−0.104	0.014
IA	−0.1311	0.04	0.001	−0.209	−0.053
PD	−0.0426	0.073	0.561	−0.187	0.101
NT	−0.0517	0.036	0.148	−0.122	0.018

**Table 4 ijerph-20-06782-t004:** Partial regression coefficients of multiple linear regression with interaction terms.

	Std. β	Std. Error	*p*-Value	95% CI	
const.	0.1277	0.092	0.163	−0.052	0.308
TE	0.5585	0.079	<0.001	0.403	0.714
ES	−0.1852	0.06	0.002	−0.303	−0.067
UE	−0.2924	0.046	<0.001	−0.383	−0.201
EW	−0.1223	0.049	0.013	−0.218	−0.026
PW	−0.0272	0.071	0.701	−0.166	0.112
IW	0.1242	0.058	0.034	0.009	0.239
CM	−0.0817	0.068	0.227	−0.214	0.051
TI [ln]	−0.0853	0.126	0.497	−0.332	0.161
FI	0.0271	0.053	0.61	−0.077	0.132
TB [ln]	−0.13	0.041	0.002	−0.21	−0.05
DS [ln]	0.0398	0.029	0.172	−0.017	0.097
CS	−0.0409	0.039	0.3	−0.118	0.036
TP	−0.0358	0.033	0.279	−0.101	0.029
IA	−0.1255	0.042	0.003	−0.208	−0.043
PD	0.2084	0.2	0.299	−0.185	0.602
NT	0.0001	0.044	0.998	−0.087	0.087
const. × trpz (PC1; −4, −0.5, 0.5, 4)	−0.3502	0.433	0.418	−1.2	0.5
TE × linz (PC1; −1.5, 0.5)	0.2429	0.166	0.144	−0.083	0.569
UE × trpz (PC1; −3, −2, 1, 3)	−0.3759	0.081	<0.001	−0.534	−0.218
EW × linz (PC1; −3, 0.5)	0.2551	0.107	0.018	0.044	0.466
PW × lins (PC1; 0, 4)	1.0704	0.69	0.121	−0.285	2.426
IW × linz (PC1; −3, 1.5)	0.3829	0.138	0.006	0.111	0.655
CM × trpz (PC1; −4, −2.5, −1, 1)	−0.2627	0.157	0.094	−0.571	0.045
TI [ln] × trpz (PC1; −2.5, −1, 1, 2.5)	−0.4119	0.138	0.003	−0.683	−0.141
TB [ln] × lins (PC1; −3, 2)	0.0501	0.108	0.642	−0.162	0.262
DS [ln] × linz (PC1; −2, 2.5)	0.1449	0.079	0.068	−0.011	0.301
CS × linz (PC1; −3.5, −0.5)	0.202	0.072	0.005	0.061	0.343
CS × lins (PC1; 2.5, 4)	0.0884	0.154	0.567	−0.215	0.392
TP × lins (PC1; −3.5, 0)	0.7658	0.521	0.142	−0.258	1.789
PD × linz (PC1; −4.5, −2)	1.3946	0.564	0.014	0.286	2.503
PD × linz (PC1; −2, 2.5)	0.3306	0.344	0.337	−0.346	1.007

## Data Availability

Publicly available datasets were analyzed in this study. The data sources were described in the Section 2.

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
