# Peer review of "Exploration of the Relationships between Men’s Healthy Life Expectancy in Japan and Regional Variables by Integrating Statistical Learning Methods"

_ijerph, 2023, doi:10.3390/ijerph20186782_

Round 1
Reviewer 1 Report
The originality of this study is quite high.
However, if the author proposed new methos in general, it should make a comparison in more generalized situation.
Similarly, the more complex the analysis, the more unstable the results tend to be given. If the authors had already evaluated it, they should describe that this method can give stable results in various types of data.
From the upper points of view, how about changing the title from the current to Exploration of municipal factors related to healthy life expectancy by combining random forest and linear regression to produce “sylvan linear-regression” for exploring the relationship between men’s healthy life expectancy in Japan and regional variables?
Reviewer 2 Report
The paper focused on developing/selecting a regression model with interaction terms for predicting the male’s life expectancy in Japan. In order to consider the paper for publication, the following issues should be addressed before the acceptance of this paper.
Major:
11. What is the sample size used in the analysis? The data used here before and after filling the missing values should be given publicly?
22. The code for reproducing the results of the proposed idea should be made public.
33. The authors used linear interpolation or carrying last observation forward for missing values. Why the appropriate imputation technique(s) is not used for imputing the missing observations. The change of results in such scenario should be considered and discussed.
44. On page 4, l136, the regression coefficients are taken from multiple linear regression fit or simple linear regression fit for each predictor separately. I believe that the considered predictors will be correlated and multiple or simple linear regression will provide different regression coefficients. Why Principal component regression is not considered to get regression coefficients.
55. Authors are considering first PC with the assumption of maximum variation covered. The usual PCs are linear combinations of all original variables. What will be the situation if the variables used for construction of PCs are not linearly related.
66. In Table 2, what is the role of given loadings. What is significance of these loadings to present? Also, these loadings are obtained with complete data with all features? What is role of this PC1 in bootstrap samples?
77. In Fig. 2, how the number of features were decided for node splitting?
88. Does Figure 2 only suggests to consider some possible interaction terms but quadratic or cubic terms also? If so, then why such functional form is not considered and tested?
99. For explanation of Figure 2, a model for one bootstrap sample should be written (it can be hypothetically/arbitrary for one bootstrap sample for understanding of reader as an example that which possible model is being used at which stage) for red dotted line, orange dashed line, and violet solid line.
110. The betas used in Figure 2 are standardized betas or they have been transformed back?
111. For the results presented in Table 3 and 4, the response and the predictors should be explicitly mentioned.
112. How the proposed method is better with respect to better prediction is not discussed. For example, if a model is considered with some features selected (without interaction terms and with first order interaction) using lasso, elastic net or any other feature selection criteria, how the predictive performance can be compared with the proposed approach.
Minor:
11. In Table 1, the text in the columns should be left align.
22. The font of labels, axis etc of Figure 2 should be increased to make it readable.
33. The p-values written as zero should be written as <0.001 throughout the text.
Minor grammatical corrections and rephrasing of sentences is required for better understanding of the proposed work.
